REGISTERED REPORT PROTOCOL

# The effect of mobile text messages on knowledge and perception towards cancer and behavioral risks among college students, Northeast Ethiopia: A randomized controlled trial protocol

**Foziya Mohammed Hussien**[1]*, **Anissa Mohammed Hassen**[2], **Zinet Abegaz Asfaw**[3], **Aragaw Yimer Ahmed**[4], **Hamid Yimam Hassen**[5]

1 Department of Public Health Nutrition, School of Public Health, College of Medicine and Health Science, Wollo University, Dessie, Ethiopia, 2 Department of Epidemiology and Biostatics, School of Public Health, College of Medicine and Health Science, Wollo University, Dessie, Ethiopia, 3 Department of Reproductive and Family Health, School of Public Health, College of Medicine and Health Science, Wollo University, Dessie, Ethiopia, 4 Department of Internal Medicine, School of Medicine, College of Medicine and Health science, Wollo University, Dessie, Ethiopia, 5 Department of Primary and Interdisciplinary Care, University of Antwerp, Antwerp, Belgium

* foziyamohammed2018@gmail.com

## Abstract

### Background

Cancer is an emerging public health problem in Ethiopia. A significant proportion of premature cancer deaths are preventable. The socioeconomic impact of cancer can be considerably reduced provided that effective interventions are put in place to control risk factors. Text-messaging has been currently targeted as a simple and efficient tool for providing people with health information.

### Objective

To evaluate the effectiveness of mobile text messages in improving knowledge and perception on cancer and behavioral risks among college students.

### Methods

*Design*: a single-blind, 2-arm randomized controlled trial will be used. *Setting*: the study will be conducted among public colleges in Dessie town, Northeast Ethiopia. *Study population*: students who are studying in all public colleges. *Intervention*: a text message related to cancer risk factors once a day for two months. Control groups will receive general health messages daily for two months. *Data*: Socioeconomic characteristics, health belief variables, and behavioral risk factors of cancer will be collected before and after the intervention. Text messages will be provided based on the health belief model (HBM). *Primary outcomes*: cancer risk knowledge score and risk perception will be measured at baseline and 6 months post-randomization. *Secondary outcomes*: a change in mean healthy diet score, physical

**Data Availability Statement:** All relevant data from this study will be made available upon study completion.

**Funding:** FMH, AMH and ZAA has received partial financial support from Wollo University for data collection and mobile voucher card. The funder had no role in the study design, data collection and analysis, decision to publish, or preparation of the manuscript. There was no additional external funding received for this study.

**Competing interests:** the authors have declared that no competing interests exist.

activity level, alcohol intake, and tobacco use will be measured at baseline-, 3, and 6 months post-randomization. ***Analysis***: We will compute descriptive statistics for each outcome pre and post-intervention. To test the change in knowledge and perception, paired t-test will be used. Analysis of Covariance will be used to test over group comparison.

## Trial registration

ClinicalTrials.gov (https://register.clinicaltrials.gov) NCT04269018.

## Introduction

Cancer is among the leading causes of premature death worldwide. In 2020, the incidence was estimated to be 19.3 million and was responsible for 10 million deaths globally [1]. Previously it was described as a disease of the high-income countries, but now it is also an important public health problem in low- and middle-income countries (LMICs) [2]. Lifestyle changes, rapid urbanization, cultural transition, and an increase in life expectancy could attribute to a rise in incidence. From 2005 to 2015, the highest (10–20%) increase in the incidence of cancer was observed in the World Health Organization (WHO) African region [3, 4]. Hence, it imposes an enormous burden on the already overwhelmed health care system of LMICs.

In Ethiopia, the numbers of cancer cases were estimated to be 77,352 and were responsible for 51,865 deaths nationally in 2020 [1]. Cancers of the breast, cervix, colorectal cancer, Non-Hodgkin lymphoma (NHL), leukemia, cancers of the prostate, thyroid, lung, stomach, and liver are the most commonly occurring cancers in the country [3–5]. Despite a rise in incidence, cancer remains among the least public health priorities in Africa, mainly because of the immense burden of communicable diseases along with limited resources [6, 7].

The risk factors for cancer can be broadly categorized into four: behavioral, biological, environmental, and genetic. Behavioral risk factors include tobacco use, harmful use of alcohol, unhealthy diet, and physical inactivity [5]. These lifestyle and environmental factors are potentially modifiable if appropriate preventive interventions are implemented [8, 9]. Human behavior is a key to the etiology of cancer and presents channels for targeted and sustained intervention [5, 10]. The effect of cancer can be significantly reduced through effective interventions to improve modifiable risk factors, early detection of cases, and appropriate management and care for those with the disease [4].

The Health Belief Model (HBM) has been one of the psychosocial approaches for explaining health related behaviors and guide the design of interventions to enhance compliance with preventive procedures [11, 12]. It has six components which originated from the belief that individuals will take action to control unhealthy behaviors if they regard themselves as susceptible to the condition; if they believe that the health problem has serious consequences; if they believe that a course of action available to them would be beneficial in reducing either susceptibility or severity of the condition; if they believe that the anticipated barriers to taking the action out-weighed by its benefits; if there is stimulus needed to trigger the decision-making process to accept a recommended health action; and if they develop confidence in their ability to successfully perform a behavior. [11, 12]. In addition, HBM states that perceptions of reality influence behavior. Health behaviors are influenced by a person's desire to avoid an unhealthy lifestyle or to get well [12, 13]. The HBM will therefore be used to develop an intervention to increase awareness of and perceived susceptibility to cancer, and affect change to four key health behaviors.

In Ethiopia cancer screening, diagnosis and management are sub-optimal and population-based data are limited to few cities [4]. Few studies in the country showed poor knowledge of risk factors and screening were identified as important factors for poor utilization of cancer screening and other prevention services [7, 14, 15]. Thus, awareness creation and behavioral intervention might address a wide variety of key processes and outcomes across the cancer control continuum from prevention to care for survivors [10]. As part of the strategy, text messages are currently considered to be the most feasible and widest-reaching mHealth intervention, as they do not require internet or any other advanced facilities [16]. Several studies indicated that text messaging interventions are effective in smoking cessation and improving physical activity [17–20]. However, the effectiveness on knowledge and behavior is not studied in the Ethiopian context. Therefore, this study aimed to measure the effectiveness of text messages to improve knowledge of behavioral risks, which could be helpful to raise cancer awareness among citizens regarding cancer issues and promote cancer prevention and control in this country. We hypothesize that mobile text messages will improve awareness and risk perception of cancer, ultimately improving risk reduction behavior. Given the demonstrated effectiveness, delivering the intervention to a larger community might significantly reduce the public health burden of this disease.

## Research questions

1. What is the effectiveness of daily mobile text messages in improving cancer risk related knowledge and perception at 6 month compared with controls?

2. What is the effectiveness of daily mobile text messages in decreasing lifestyle cancer risk factors including physical inactivity, unhealthy dietary habit, smoking, and alcohol consumption at 6 month?

## Trial objectives

Our study objective is to assess the effect of cancer risk specific daily mobile text messages, for early adults, on cancer risk knowledge, perception and risk reduction behaviors such as healthy dietary habit, physical activity, reduction of alcohol consumption, and quit smoking, compared to a general health message once a week.

## Methods

### Study design

The study will be a single-blinded, parallel-group, randomized controlled trial, which aims to recruit college students in Northeast Ethiopia. It is designed to allow for a before-after comparison of cancer risk knowledge, perception, and behavior between an intervention and a control group. The main reason to conduct a randomized study is to control for some of the important confounding variables. Once the most important variables are being adjusted, the rest residual confounders will be expected to be controlled in the randomization. The CONSORT statement was used in the design of this study and will be employed in reporting of findings [21].

### Study setting

The study will be conducted in all public colleges in Dessie town, the administrative town of South Wollo, Northeast Ethiopia. It is 401 kilometers far from Addis Ababa, the capital city of Ethiopia. There are three public colleges in the city, which provides several academic and training services. All public colleges in the city will be included, namely Woizero Siheen

Polytechnic College, Dessie College of Teachers Education, and Dessie Health Science college. Woizero Siheen Polytechnic College was established in 1930 and currently provides technical and vocational education and training (TVET) in 5 different campuses: Main Campus (Siheen Campus), Merho Campus, Hotie Campus, Dawdo Campus, and Menen (Vocational Campus). The College has 5,975 trainees in regular and night-shift programs. Dessie College of teacher's education was established in 1980 and has around 1,426 students. Dessie Health Science College will not be included, as the students in this college are expected to have a higher knowledge on risks of cancer and preventive mechanisms.

## Eligibility criteria

Participant's eligibility criteria include: 1) being a student in selected public colleges; 2) aged 18 to 35 years; 3) No prior diagnosis of any type of cancer; 4) access to mobile text messaging; 5) not a student of any health science college; 7) able to provide informed consent, and 8) not enrolled in another interventional study that uses text message as a tool.

## Sample size and sampling strategy

The sample size for the study was determined using the assumption of a design to demonstrate the superiority of a new intervention or text message compared to the control group in improving self-efficacy (superiority trial design), assuming a 2.9 mean difference of perceived self-efficacy to adopt healthy behavior between text message and control group, a 6.22 pooled standard deviation [22], 95% confidence level, 80% power, and an expectation of 10% loss to follow up. With a 1:1 ratio, the sample size was estimated to be 160 (80 intervention and 80 control group). The sample size will be allocated to Woizero Siheen Polytechnic College and Dessie College of teacher's education based on proportional to the number of students. Students' ID numbers will be obtained from the registrar's office of each college and used as a sampling frame. Then; a simple random sampling method will be used to select study subjects.

## Recruitment and randomization

Potential participants will be recruited from selected public colleges. After we gained an official letter from Wollo University, each college's administrator and the registrar will be communicated to confirm their willingness and obtain a student list with ID number. Selected students will be requested for consent to participate in the study, and giving their phone number for delivering text messages and subsequent evaluation. After the baseline assessment, eligible participants will be randomized to either the intervention group (a daily cancer risk specific text messages) or control (a daily general health text messages) in a uniform 1:1 allocation ratio, using a computer-based randomization program. Allocation concealment will be done using SNOSE (sequentially numbered opaque, sealed envelope) technique when the students are enrolled in the study. The students' ID no. will be put in an envelope, then the investigators will randomize the intervention group accordingly. The computer-generated sequence will be performed by a statistician, independent of the investigators (Fig 1). After the intervention, we will conduct a panel discussion (to appreciate for successful completion of the intervention and provide an appointment for end line assessment) with students in a separate room for cases and controls. Four months back after the intervention, they will receive a phone call for end line assessment.

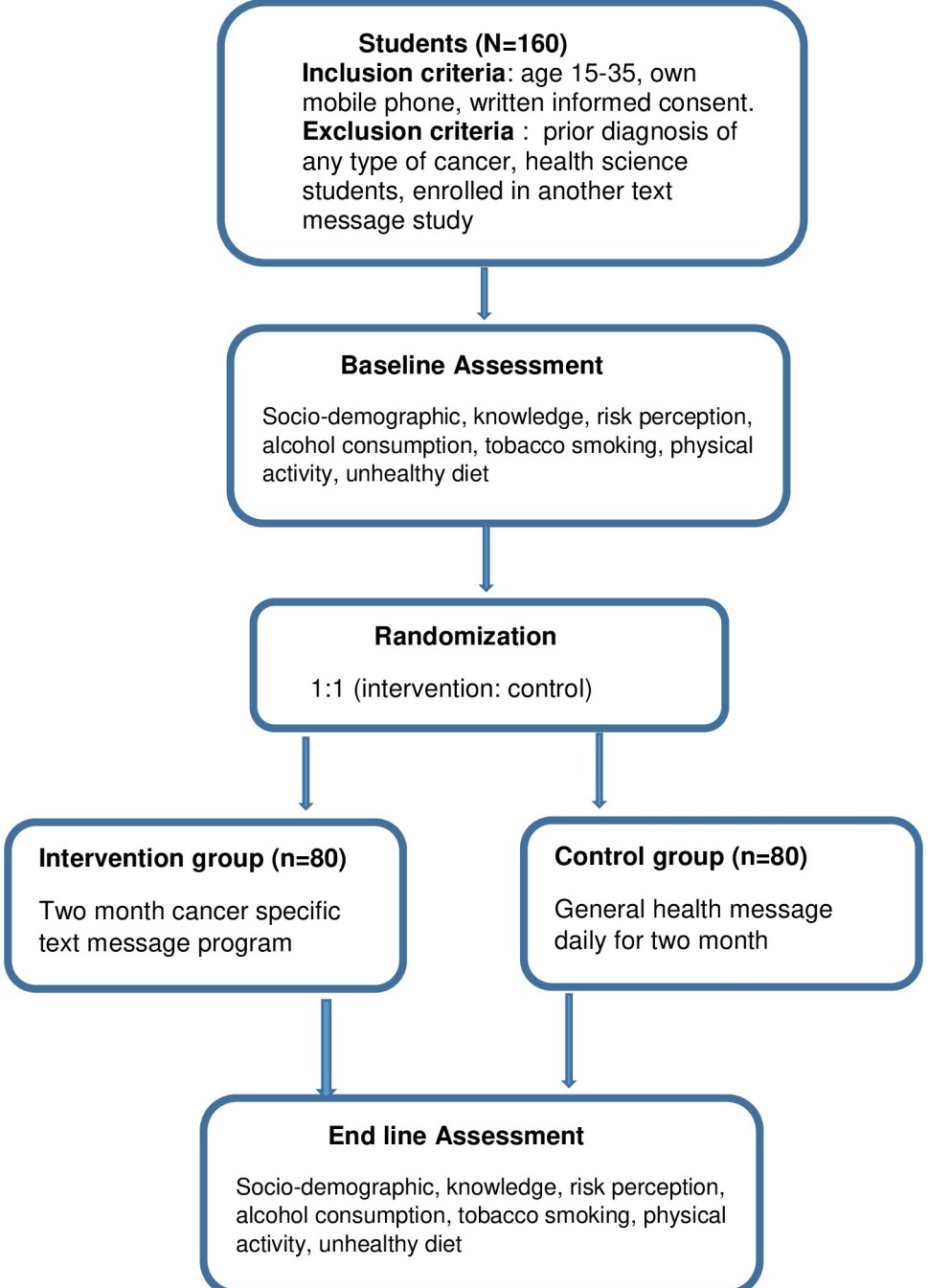

**Fig 1. Study flow diagram of text message program on knowledge and perception towards behavioral risk factor of cancer.**

## Blinding

To maintain blinding, a welcome and general health mobile text message will be delivered to control participants. Moreover, the participants will not be informed whether they are in the intervention or control arm.

## Intervention

Participants randomized into the intervention arm will receive a daily mobile text message for 2 months. The text messages will be sent to participants in the morning. The principal investigator will be responsible to check the status of each sent item. If a message fails to be delivered, it will be sent again to the recipient. If delivery status keeps failing, a phone call will be performed to discuss the issue and take measures accordingly.

The messages will be prepared and delivered based on the health belief model (HBM). The focus of text messages will be on the prevention of behavioral risk factors of cancer such as harmful use of alcohol, unhealthy diet, tobacco smoking, and physical inactivity. Here are some examples of text messages: "although none of your family members suffers from cancer, there is still a chance of developing cancer; having unhealthy diet increases the risk of cancer, so try to improve healthy food habits; in case you are forced to consume commercially produced foods, carefully read the tag and buy only the foods which include the lower amount of additives and preservatives"; "quit smoking is one of the preventive ways of cancer"; "alcohol can cause diseases like cancer, consider setting a goal to reduce drinking step by step and quit it through time"; "being physically active is critical for well-being and to prevent cancer, it can help to add years in your life"; "performing at least 30 minutes of physical activity per day helps to prevent cancer"; "be proud of yourself for having healthy lifestyles to prevent cancer!". Whereas participants randomized to the control group will receive an initial welcome text message similar to the intervention group. Then, they will receive a general health message (not cancer-specific and not related to the four lifestyle risks) daily for two months (e.g. wash your hand with soap after toilet and before each meal will decrease the risk of diarrhea, etc.) (Table 1).

## Outcome measures

All the primary and secondary outcomes will be measured at baseline and 6 months post-randomization. A self-administered questionnaire will be used to collect the baseline and post-intervention socioeconomic characteristics, knowledge towards cancer, risk perception, and lifestyle behaviors.

**Table 1. Sample of text messages with each health belief model variables for evaluating text message intervention on the behavioral risks of cancer, Ethiopia, 2021.**

| Health belief model variables | Text messages |
|---|---|
| Perceived severity | Cancer is with severe pain and discomfort |
| Perceived susceptibility | All of us are faced with the danger of cancer, so we must think of the prevention |
| | Although none of your family members suffers from cancer, there is still a chance of developing cancer |
| | Having unhealthy diet increases the risk of cancer, so try to improve healthy food habits |
| Perceived barriers | in case you are forced to consume commercially produced foods, carefully read the tag and buy only the foods which include the lower amount of additives and preservatives |
| Perceived benefit | Try steaming and baking to reduce the need for excess oil when cooking |
| | Quit smoking is one of the preventive ways of cancer |
| | Being physically active is critical for well-being and to prevent cancer, it can help to add years in your life |
| | Performing at least 30 minutes of physical activity per day helps to prevent cancer |
| Self-efficacy | Keep pushing forward toward a smoke-free and healthier life |
| Cues to action | Be proud of yourself for having healthy lifestyles to prevent cancer |

**Primary outcomes.** The primary outcomes for this trial are knowledge of behavioral risks of cancer and risk perception. Knowledge will be assessed with a series of questions regarding high-risk groups, risk factors, complications, and preventive mechanisms of cancer with "yes", "no", and "I do not know" options. To assess the risk perception of cancer, we will obtain data on the health belief model variables: Perceived susceptibility towards cancer (e.g. You perceived that you will get risk of developing cancer at any time of your life), Perceived severity of cancer (e.g. "Cancer is fatal, and its complication are dangerous/severe in life") Perceived benefits of adopting behaviors (e.g. "Having a healthy nutrition would decrease the probability of any type of cancers") Perceived barriers to adopting behaviors (e.g. "My monthly income is insufficient to take the recommended diet for the prevention of cancer"), Perceived self-efficacy for adopting dietary behaviors (e.g. "I am confident that I can prevent cancer through healthy lifestyle") Internal cues for adopting dietary behaviors (e.g. "when knowing the death of any by cancer, it flips me to perform the preventive behaviors") the items of this subscale will be measured on a Likert scale ranging from 1 = "Strongly disagree" to 5 = "Strongly agree" [22].

**Secondary outcomes.** *Dietary habit.* will be measured using a food frequency questionnaire adapted from the dietary assessment primer which was developed by the USA national cancer institute and then modified based on the study objectives and local staple foods. In the questionnaire, the type, and frequency of food items a participant consumed in the past 1 month will be assessed [23].

*Physical activity level.* will be measured using the International Physical Activity Questionnaires short form (IPAQ-S). The questions will ask the number of days and time participants spent doing vigorous physical activity, moderate physical activity, walking, and sitting in the last 7 days [24].

*Alcohol intake.* will be assessed using AUDIT a tool adapted from World Health Organization (WHO) which we will use to assess participant's consumption in the past 1 month [25].

*Tobacco use.* will be assessed using a series of questions adapted from the World Health Organization (WHO), which will assess the duration, quantity, and frequency of smoking [26].

## Data quality control and management

The validity of the items was checked by an expert panel of specialists in health education and nutrition. They provided intellectual judgment on the necessity and relevance of the scale items. The questionnaire was first prepared in English language and translated to Amharic and again re-translated into English by another person to check for consistency. About 10% of the questionnaire was pretested in Logo Hayiq Polytechnic College and then internal consistency was checked until the acceptable level was reached. To control the data quality, all data collectors will be trained for two days on the aim of the study, methods, and how to take informed consent and administer the questionnaire. Supervisors will monitor the data collection process and the principal investigator will visit when needed. The collected data will be checked for completeness and consistency, then categorized, coded, and entered using Epidata version 3.1 (https://epidata-entry.software.informer.com/3.1/). Then, the clean data will be exported for analysis. Data access will be limited to authorize persons only. The questionnaires will be stored and preserved for future reference.

## Statistical analysis

A descriptive summary such as; frequencies, proportions, and cross-tabulations will be used to present categorical variables. For continuous variables, normality of the data will be checked

then will be summarized either using mean with standard deviation or median with interquartile range. The primary and secondary outcomes will be analyzed based on the intention-to-treat approach. To evaluate whether the randomization results compare groups, we will compare for differences in participants' baseline characteristics between two groups using a two-sample *t*-test or non-parametric tests for continuous outcomes and by a Chi-square test, Fisher's exact, or non-parametric tests for categorical outcomes. To assess the change in intention to quit smoking or reduce alcohol intake, we will use chi-square test or any other non-parametric tests depending on the assumptions. Student's paired samples t-test will be used to test within-group changes in terms of HBM variables and behavioural outcomes pre and post-intervention. The analysis of covariance will be used to make over-group comparisons. To estimate the population-averaged effect of the intervention on outcome measures, we will develop a marginal model using multivariable generalized estimating equations (GEEs). Data will be reported as mean ± SD. The significance level for all of the results will be set at the $P<0.05$ level.

## Ethical approval and consent to participate

Ethical approval for the research was obtained from Wollo University, College of Medicine and Health Sciences Research Ethics Review Committee. Written informed consent will be obtained from participants after a detailed explanation of the study purpose, the benefits, and the risks of participation. Privacy and confidentiality of the collected information will be ensured throughout the process; measures will be taken to ensure respect, dignity, and freedom of each participant in the study. All information gained during the study will be kept strictly confidential. Data will be stored in a password-protected database.

## Discussion

In low- and middle-income countries, policies and programs mainly focus on communicable diseases. However, recently due to lifestyle change, cancer and other non-communicable diseases are emerging leading to a double burden in these countries. Most of these diseases can be prevented through targeted behavioral intervention. Considering the extent of the problem and scarcity of resources, it is imperative to provide a feasible and scalable primary prevention strategy to enable policymakers to make appropriate choices and facilitate implementation.

This simple and cost-effective text message intervention uses positively-framed, informative, and semi-personalized messages to motivate students for maintaining a healthy lifestyle. Moreover, this intervention package particularly focuses on improving the main modifiable risk factors including dietary habits, physical activity, alcohol intake, and tobacco use. Having good knowledge of risk factors and risk perception towards cancer is an integral part of cancer risk reduction. It would help those with unhealthy lifestyles to change and those with a healthy lifestyle to persist in that behavior.

The results from this intervention study will also be used to scale up the strategy to the development of mobile applications equipped with texts and short briefs on cancer risk and prevention. This strategy will help public health professionals and advocates to reach a wide range of audiences with less cost, less time, and without a need to travel long. The application will incorporate the up-to-date health information technology system to improve communication in the area of creating awareness and dissemination of information to prevent cancer at the community level. Information regarding common risk factors like smoking, alcohol drinking, obesity, physical inactivity, unhealthy diet, and foods recommended for the prevention of cancer will be included.

If successful, this study will inform a larger multicenter RCT, to scale up the intervention to the wider community to reduce the preventable causes of cancer. It will also be relevant for policymakers and other stakeholders at various levels to implement the intervention program to a larger extent, with the ultimate goal of decreasing the burden of cancer in the study area as well as in the country.

Texting message intervention as a technology can have more reach than newer technologies (e.g. smartphones app) as they can be delivered on any mobile device with no internet. However, the SMS texts can be ignored, the options for content are very limited, and there is little or no possibility to assess engagement with the texts objectively. Moreover, for a text message intervention participants should be able to read and write and have basics of mobile use. Hence, we will assess the feasibility and practicality to scale up this intervention to the general population in the future if it is effective among college students.

## Supporting information

**S1 Checklist. CONSORT 2010 checklist of information to include when reporting a randomized trial.**
(DOC)

**S2 Checklist.**
(DOC)

**S1 File.**
(DOCX)

## Acknowledgments

We are grateful to Wollo University research coordinating office for allowing us to prepare this research protocol.

## Author Contributions

**Conceptualization:** Foziya Mohammed Hussien.

**Data curation:** Foziya Mohammed Hussien, Aragaw Yimer Ahmed.

**Formal analysis:** Hamid Yimam Hassen.

**Funding acquisition:** Foziya Mohammed Hussien, Anissa Mohammed Hassen, Zinet Abegaz Asfaw.

**Investigation:** Foziya Mohammed Hussien, Aragaw Yimer Ahmed.

**Methodology:** Foziya Mohammed Hussien, Hamid Yimam Hassen.

**Project administration:** Anissa Mohammed Hassen, Zinet Abegaz Asfaw.

**Resources:** Anissa Mohammed Hassen, Zinet Abegaz Asfaw.

**Software:** Hamid Yimam Hassen.

**Supervision:** Foziya Mohammed Hussien, Aragaw Yimer Ahmed, Hamid Yimam Hassen.

**Validation:** Aragaw Yimer Ahmed, Hamid Yimam Hassen.

**Visualization:** Hamid Yimam Hassen.

**Writing – original draft:** Foziya Mohammed Hussien.

**Writing – review & editing:** Foziya Mohammed Hussien, Anissa Mohammed Hassen, Zinet Abegaz Asfaw, Aragaw Yimer Ahmed, Hamid Yimam Hassen.

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
