## [Decision Letter · Decision Letter 0]

28 Jul 2020

PONE-D-20-05561

The effect of mobile text message on behavioral risk factors of cancer among college students, northeast Ethiopia: a randomized controlled trial protocol

PLOS ONE

Dear Dr. Hussien,

Thank you for submitting your manuscript to PLOS ONE. After careful consideration, we feel that it has merit but does not fully meet PLOS ONE’s publication criteria as it currently stands. Therefore, we invite you to submit a revised version of the manuscript that addresses the points raised during the review process.

Please provide thorough responses to the reviewers' comments, particularly to the questions posed by Reviewer 1 and Reviewer 3.

We look forward to receiving your revised manuscript.

Kind regards,

Nancy Beam, PhD

Staff Editor

PLOS ONE

Journal Requirements:

Additional Editor Comments (if provided):

Reviewers' comments:

Reviewer's Responses to Questions

**Comments to the Author**

1. Does the manuscript provide a valid rationale for the proposed study, with clearly identified and justified research questions?

Reviewer #1: Yes

Reviewer #2: Yes

Reviewer #3: Yes

2. Is the protocol technically sound and planned in a manner that will lead to a meaningful outcome and allow testing the stated hypotheses?

Reviewer #1: Partly

Reviewer #2: Yes

Reviewer #3: Yes

3. Is the methodology feasible and described in sufficient detail to allow the work to be replicable?

Reviewer #1: Yes

Reviewer #2: Yes

Reviewer #3: Yes

4. Have the authors described where all data underlying the findings will be made available when the study is complete?

Reviewer #1: Yes

Reviewer #2: Yes

Reviewer #3: Yes

5. Is the manuscript presented in an intelligible fashion and written in standard English?

Reviewer #1: Yes

Reviewer #2: Yes

Reviewer #3: Yes

6. Review Comments to the Author

You may also provide optional suggestions and comments to authors that they might find helpful in planning their study.

Reviewer #1: This is a "Registered Report Protocol", where the investigative team plans to conduct a 2-arm randomized controlled trial (RCT) to test the effectiveness of mobile text messages in providing necessary health information for cancer prevention, compared to controls. While the study design is appropriate, I have some serious concerns on the actual science behind the study. In particular, I am not sure what an over-simplified RCT of this sort (spanning across only 6 months) will actually do, if any, to the cause. The intervention will be delivered, strictly, via text messages, to non-cancer (or, no prior diagnosis of cancer) subjects. There is no way to ascertain even whether the subjects are "actually reading/attending" the text messages, and responding to those (given that none of the subjects enrolled are actually cancer patients). Furthermore, following those for 6-months seemed to be a very short time to evaluate any effectiveness in that regard. In my opinion, all these will entail a "very pilot" study.

I do have some additional comments that require attention.

1. Sample size/Power: The sample size/power statement should explicitly mention the statistical test used.

2. Randomization: From the randomization procedure described, it is not clear if this is a simple randomization, or a block. I would recommend a block strategy, which will ensure sample size are balanced across groups over time.

3. Randomization: The authors need to mention how "Allocation Concealment", and "Blinding", considered two separate concepts, will be actually performed. Refer to this article below:

https://www.bmj.com/content/347/bmj.f5518

4. Statistical Analysis plan: The design is longitudinal; so I was wondering why a small longitudinal analysis plan was not proposed. A longitudinal analysis (via mixed models, or generalized estimation equations) will allow one to have a better picture on how the trial actual progressed, and the subsequent estimates of the relevant parameters.

Reviewer #2: Dear Author,

First of all, I appreciate that this is a great effort and I wish successful completion in the implementation of this clinical trial!

Follows, some general comments, and afterwards I suggest some corrections in some small syntax or logical errors:

• The bibliographic reporting system is incomplete. There is no reference within the text to the cited bibliography at the end.

• in page 1: Department of epidemiology and biostatics … it’ ok… but it would be better to use the word "biostatistics". This term is derived from the Greek term “βιοστατιστική” which is pronounced “biostatistiki”.

• in page 8: Epidata version 3.1: There is no relevant link. Moreover, it would be helpful to have a brief explanation, as a footnote, about using this software.

Also, please correct:

• in page 1: “Department of public health nutrition, ,…” with: “Department of public health nutrition, ….”

• in page 4: “..which could be helpful to raise cancer awareness…” with: “..which could be helpful to raise awareness among citizens regarding cancer issues..”

best regards

good luck

Reviewer #3: The goal of the proposed study is to evaluate the effectiveness of mobile text message in improving behavioral risks of cancer among college students in Ethiopia. The proposed study addresses an important topic and has the potential to be an effective intervention aimed at improving cancer risk knowledge and perception.

The authors plan to send a text message once a day for two months, which seems feasible. The daily messaging strategy should be effective given the relevant literature, but have the researchers done any kind of pilot testing to determine the appropriate amount of text messages for college students? They might become fatigued after receiving a message everyday for two months and tune out the messages, especially since those in the treatment group are receiving messages about one topic (cancer risk).

Why are the participants in the control group only receiving one message per week and not one a day? Can the researchers also explain their rationale for measuring cancer risk knowledge and risk perception at baseline and six months post-randomization? Why not immediately after the two months?

On page three, the authors state, “ Behavioral risk factors include tobacco use, harmful use of alcohol, unhealthy diet and physical inactivity [4]. These lifestyle factors are potentially avoidable if appropriate preventive interventions are implemented.” I don’t disagree with this claim, but the authors should also consider the structural factors and a lack of access to resources in some cases that can prevent one from making lifestyle changes.

The health belief model is mentioned at various points throughout the manuscript, but there is no background information presented in the literature review. There should also be a stronger rationale for why this framework was chosen for the current study.

There needs to be more information provided in the “recruitment and randomization” section. Based on the information here, it is difficult to determine if this study is feasible. Have the administrators from the universities already agreed to these protocols? Are there contracts in place with a service provider or third party who will be sending the messages? Who will write the messages? How will students opt in to the program? What is the procedure for opting out? How will students provide consent over text messaging?

There also needs to be additional details provided about the messages. Are the researchers writing the messages? Are staff and faculty from the universities writing them? Will there be student/peer input so the language resonates with the college students? Will a messaging expert review these before they are sent out? It would also be helpful to see examples of the messages the researchers plan to use.

“Primary outcomes” section, page seven- please cite the scales you plan to use to measure the health belief model variables.

Will the surveys be sent over the phones?

The NIAAA measure for alcohol intake will be used to “assess participants’ consumption in the past 1 month.” However, this does not seem to align with the original timeline for collecting participant data.

On page eight, the authors make it sound as though the questionnaires will be administered in person. Is this the case? If so, how will responses be matched (baseline and post-test) and how will the researchers match responses with phone numbers/text message (control or treatment) group? There needs to be more details about how responses to the questionnaire items (pre and post) will be matched with participants in the text messaging groups. Also, can the authors explain why they are administering the questionnaires in person rather than through a survey link sent in a text message?

I would also like to see more information about the message content. The researchers mention on page nine that the messages will be “positively-framed” and “semi-personalized.” Can the researchers provide examples of these? Is there a communication specialist on this project who is writing and/or reviewing the messages in accordance with communication theory?

7. PLOS authors have the option to publish the peer review history of their article (what does this mean?). If published, this will include your full peer review and any attached files.

Reviewer #1: No

Reviewer #2: No

Reviewer #3: No

---

## [Author Response · Author response to Decision Letter 0]

18 Sep 2020

Reviewer #1:

This is a "Registered Report Protocol", where the investigative team plans to conduct a 2-arm randomized controlled trial (RCT) to test the effectiveness of mobile text messages in providing necessary health information for cancer prevention, compared to controls. While the study design is appropriate, I have some serious concerns on the actual science behind the study. In particular, I am not sure what an over-simplified RCT of this sort (spanning across only 6 months) will actually do, if any, to the cause. The intervention will be delivered, strictly, via text messages, to non-cancer (or, no prior diagnosis of cancer) subjects. There is no way to ascertain even whether the subjects are "actually reading/attending" the text messages, and responding to those (given that none of the subjects enrolled are actually cancer patients). Furthermore, following those for 6-months seemed to be a very short time to evaluate any effectiveness in that regard. In my opinion, all these will entail a "very pilot" study.

• Response: Thank you for your comment. The aim of the study is to test the effect of text message on changing knowledge, perception and behavior of unhealthy lifestyle which led to cancer. We agree that, as a digital public health intervention, it is not straightforward to ascertain whether participants read it or not. We can confirm whether the text is delivered, using the delivery report. We planned to assess the effectiveness under the assumption of the participants will read the text message, means, it is also part of the effectiveness. We also agree that 6 months may not be sufficient to evaluate effectiveness for a large scale intervention. However, this study will provide an indication whether the intervention has a potential for large scale intervention. Therefore, the study is a pilot effectiveness study. Based on the result we will scale-up to a large area with longer follow-up. Actually, 6 month could be sufficient to assess the change in knowledge and perception. However, the effectiveness on behavior change needs longer follow up period. If the perception is changed, we also expect a change in short term behavioral change. Hence, given the feasibility, the current protocol will pilot the effectiveness within 6 month.

Sample size/Power: The sample size/power statement should explicitly mention the statistical test used.

• Response: Thank you for the comment. The statistical test to be used is mentioned in the data analysis plan in line 215-230. The sample size with assumed power is described in line 131 of the revised version. 

Randomization: From the randomization procedure described, it is not clear if this is a simple randomization, or a block. I would recommend a block strategy, which will ensure sample size are balanced across groups over time.

Response: Thank you for your suggestion. We will use a block randomization and we described it in line number 105. 

Randomization: The authors need to mention how "Allocation Concealment", and "Blinding", considered two separate concepts, will be actually performed.

• Response: Blinding: All students will be told that, they will receive a text message from a specific mobile number and will not be told whether they are in the intervention or control group. Then a general health text message will be sent to the control group and the intervention text will be sent to the intervention group 

• Allocation concealment: this will be done using SNOSE (sequentially numbered opaque, sealed envelope) technique when the students are enrolled in the study. The students’ ID will be put in envelop, then the investigators will randomize the intervention group accordingly.

Statistical Analysis plan: The design is longitudinal; so I was wondering why a small longitudinal analysis plan was not proposed. A longitudinal analysis (via mixed models, or generalized estimation equations) will allow one to have a better picture on how the trial actual progressed, and the subsequent estimates of the relevant parameters.

Response: Thanks for your comment. Given the randomization is appropriate, simple statistical tests could be sufficient. However, it is also possible to consider the dependency in measurements before and after the intervention. We accept your comments and we revised our analysis plan in line 215-230.

Reviewer #2:

• The bibliographic reporting system is incomplete. There is no reference within the text to the cited bibliography at the end.

Response: Thank you for your comment. We revised it accordingly

• In page 1: Department of epidemiology and biostatics … it’ ok… but it would be better to use the word biostatistics

Response: Thank you for your comment we have corrected it as suggested.

• In page 8: Epidata version 3.1: There is no relevant link. Moreover, it would be helpful to have a brief explanation, as a footnote, about using this software.

Response: We added the relevant link.

• in page 1: “Department of public health nutrition, ,…” with: “Department of public health nutrition, ….”

Response: We corrected it as suggested 

• in page 4: “..which could be helpful to raise cancer awareness…” with: “..which could be helpful to raise awareness among citizens regarding cancer issues..”

Response: We modified it as suggested 

Reviewer #3:

1. The authors plan to send a text message once a day for two months, which seems feasible. The daily messaging strategy should be effective given the relevant literature, but have the researchers done any kind of pilot testing to determine the appropriate amount of text messages for college students? They might become fatigued after receiving a message every day for two months and tune out the messages, especially since those in the treatment group are receiving messages about one topic (cancer risk).

Response: Thank you for your comment. It is true that the participants might be fatigued. To minimize it, we will frame the text messages to be brief, easy to understand and attractive.

2. Why are the participants in the control group only receiving one message per week and not one a day? Can the researchers also explain their rationale for measuring cancer risk knowledge and risk perception at baseline and six months post-randomization? Why not immediately after the two months?

Response: Thank you for your suggestion. The main reason to send once a week is feasibility reason. We will consider sending other healthy related messages for control groups daily for two month. It is possible to measure knowledge and perception at two month however, as the follow up duration is short, it will overestimate the effectiveness. 

3. On page three, the authors state, “Behavioral risk factors include tobacco use, harmful use of alcohol, unhealthy diet and physical inactivity [4]. These lifestyle factors are potentially avoidable if appropriate preventive interventions are implemented.” I don’t disagree with this claim, but the authors should also consider the structural factors and a lack of access to resources in some cases that can prevent one from making lifestyle changes.

Response: Thank you for your suggestion. We agreed with your idea it is important to add such factors if feasible. However, in low-income countries, persons with full resource access could mostly practice unhealthy lifestyle because they can easily purchase tobacco, alcohol, frequently consume unhealthy diet (fried foods, meat, cream, butter, cake this are costly foods) and practice sedentary life style (no walk always use car for transportation). This all are considered as the manifestation of rich person. When we come up with our concern this factors are potentially avoidable if they know such factors lead to fatal outcome, even the poorer can also minimize these lifestyle factors.

4. The health belief model is mentioned at various points throughout the manuscript, but there is no background information presented in the literature review. There should also be a stronger rationale for why this framework was chosen for the current study.

Response: Thank you for your comment. We added a paragraph in the introduction section from line 77 to 82. The HBM is one of the most widely used theoretical frameworks for understanding health behavior. It is psychosocial model that is designed to help understand health behaviors which prevent disease, or detect disease when a patient has little, or no symptoms. Unlike other models used to describe and predict health behavior, such as theory of planned behavior, the HBM focuses on intra-personal factors, including risk related beliefs which influence individuals’ health related decision making. This is particularly important for this study as it aims to determine knowledge and risk perception of cancer. 

5. There needs to be more information provided in the “recruitment and randomization” section. Based on the information here, it is difficult to determine if this study is feasible. Have the administrators from the universities already agreed to these protocols? Are there contracts in place with a service provider or third party who will be sending the messages? Who will write the messages? How will students opt in to the program? What is the procedure for opting out? How will students provide consent over text messaging?

Response: We added some sentences in “recruitment and randomization” section as suggested from line 144 to 154. We already agreed with each college officials. During baseline assessment we will receive each student’s phone number with ID and inform that they will receive text message in the coming two months. We will submit text messages to Wollo University research coordinating office who approved our protocol then they will send to students. 

6. There also needs to be additional details provided about the messages. Are the researchers writing the messages? Are staff and faculty from the universities writing them? Will there be student/peer input so the language resonates with the college students? Will a messaging expert review these before they are sent out? It would also be helpful to see examples of the messages the researchers plan to use.

Response: Thank you for your suggestion. We revised it accordingly. The intervention message will be developed by a multidisciplinary team of oncologists, students from the college, nutritionist, and psychologists in iterative way. The message will be developed based on the American cancer society guideline for Diet and physical activity and SEOM clinical guidelines to primary prevention of cancer. Then, health education expert and nutrition specialist will revise messages, we mentioned it in “data quality control and management section” line 203 and 204. Finally, Wollo University research coordinating office will send messages with the supervision of investigators. When the program started, they will receive a phone call to inform that the message is delivering to them starting from now on wards. Daily, we will check delivery report and will assure that who will receive messages, if the participants not received messages they will receive phone call and resend it again. We add one example of text message in the “intervention section” line number 169 and 170. 

7. “Primary outcomes” section, page seven- please cite the scales you plan to use to measure the health belief model variables.

Response: Thank you for your suggestion. We revised it as suggested. 

8. Will the surveys be sent over the phones?

Response: Yes 

The NIAAA measure for alcohol intake will be used to “assess participants’ consumption in the past 1 month.” However, this does not seem to align with the original timeline for collecting participant data.

Response: Thank you for your comment. One month is the time frame for the NIAAA, which is a standard tool. Both pre- and post- intervention measures will consider past one month. It is not align but we will measure its effect as one variable. Moreover, it is a standard tool which used to capture data on usual alcohol consumption unless we don’t know the participant take alcohol in the occasion. 

9. On page eight, the authors make it sound as though the questionnaires will be administered in person. Is this the case? If so, how will responses be matched (baseline and post-test) and how will the researchers match responses with phone numbers/text message (control or treatment) group? There needs to be more details about how responses to the questionnaire items (pre and post) will be matched with participants in the text messaging groups. Also, can the authors explain why they are administering the questionnaires in person rather than through a survey link sent in a text message?

Response: We will measure it as a group as we will infer it to the population, and not concern the individual data which match at the baseline and the end. We will conduct paired t test to measure with in group variation and ANCOVA for over group variation. 

The questionnaire include sensitive question in which the participant don’t disclose unless self-administered. Moreover, it is a part of ethics to maintain the participant’s privacy in providing unbiased data.

10. I would also like to see more information about the message content. The researchers mention on page nine that the messages will be “positively-framed” and “semi-personalized.” Can the researchers provide examples of these? Is there a communication specialist on this project who is writing and/or reviewing the messages in accordance with communication theory?

Response: health education expert and nutrition specialist will revise messages, we mentioned it in “data quality control and management section” line 203 and 204. We add one example of text message in the “intervention section”. (E.g. having unhealthy diet increases the risk of cancer in each person, so unhealthy food habits should be put aside) line number 169 and 170.

---

## [Decision Letter · Decision Letter 1]

26 Nov 2020

PONE-D-20-05561R1

The effect of mobile text message on behavioral risk factors of cancer among college students, Northeast Ethiopia: a randomized controlled trial protocol

PLOS ONE

Dear Dr. Hussien,

Thank you for submitting your manuscript to PLOS ONE and for responding to the previous round of reviewers' comments. There remain a number of points which have not been fully addressed, as explained in reviewer 3's comments appended below.

Therefore, we invite you to submit a revised version of the manuscript that addresses these additional points raised during the review process.

I would also suggest that you change 'text message' to 'text messages' in your title, as that is more grammatical.

We look forward to receiving your revised manuscript.

Kind regards,

Dr Joseph Donlan

Staff Editor

PLOS ONE

Reviewers' comments:

Reviewer's Responses to Questions

**Comments to the Author**

1. Does the manuscript provide a valid rationale for the proposed study, with clearly identified and justified research questions?

Reviewer #1: Yes

Reviewer #3: Yes

2. Is the protocol technically sound and planned in a manner that will lead to a meaningful outcome and allow testing the stated hypotheses?

Reviewer #1: Yes

Reviewer #3: Yes

3. Is the methodology feasible and described in sufficient detail to allow the work to be replicable?

Reviewer #1: Yes

Reviewer #3: Yes

4. Have the authors described where all data underlying the findings will be made available when the study is complete?

Reviewer #1: Yes

Reviewer #3: Yes

5. Is the manuscript presented in an intelligible fashion and written in standard English?

Reviewer #1: Yes

Reviewer #3: Yes

6. Review Comments to the Author

You may also provide optional suggestions and comments to authors that they might find helpful in planning their study.

Reviewer #1: The authors addressed my previous comments to a greater degree of satisfaction. I have no further comments.

Reviewer #3: Thank you to the authors for taking the time to provide thoughtful responses to my comments.

The authors did not fully respond to my first question. Did they complete a pilot study or collect some kind of preliminary data to determine the amount of messages to send? For this same point, can the authors explain what they mean by framing messages so they are "attractive"?

The authors did not do a sufficient job of elaborating on the HBM in the literature review. There is no explanation of the major constructs including barriers, benefits, efficacy, cues to action.

The inclusion criteria states that participants must be between 15-35. Can the authors provide a rationale for this large age gap? This is a wide range and the health behaviors of adults in their 20s/30s are likely to be different from individuals in their teens.

The authors did include a sample message, but I would like to see more examples of the content that will be used in this study. Given that this is a messaging study, there needs to be more explanation of the actual messages and what they look like.

7. PLOS authors have the option to publish the peer review history of their article (what does this mean?). If published, this will include your full peer review and any attached files.

Reviewer #1: No

Reviewer #3: No

---

## [Author Response · Author response to Decision Letter 1]

1 Dec 2020

PONE-D-20-05561

The effect of mobile text message on behavioral risk factors of cancer among college students, northeast Ethiopia: a randomized controlled trial protocol

PLOS ONE

On behalf of all authors, I would like to thank editors and reviewers for their continued constructive comments and suggestions to improve the excellence of this manuscript. We revised the manuscript based on the comments. We hope that we addressed all the comments and questions. Here is a point by point response for each comment and question.

Editor:

I would also suggest that you change 'text message' to 'text messages' in your title, as that is more grammatical.

Response: Thank you for your comment we modified it as suggested 

Reviewer #3:

1. The authors did not fully respond to my first question. Did they complete a pilot study or collect some kind of preliminary data to determine the amount of messages to send? For this same point, can the authors explain what they mean by framing messages so they are "attractive"? 

Response: Thank you for your comment. We agree that the volume and the content of text messages should be piloted. We developed some messages, but they need to be pre-tested before applying to the main trial. We will conduct a pilot test of messages in a similar group of population at Haiq Polytechnic College, a city 40 km far from Dessie. During the pilot test we will evaluate whether the messages are easily understandable, the amount and volume of the messages through discussion with the pilot participants and measures will be taken accordingly for the main trial. We said the messages ‘attractive’ means they should be good-looking and appealing. If text messages are framed in a wrong way, they will not deliver the correct content. 

2. The authors did not do a sufficient job of elaborating on the HBM in the literature review. There is no explanation of the major constructs including barriers, benefits, efficacy, cues to action.

Response: Thank you for your suggestion. We added more descriptions on the introduction section from line 75 to 83 which explains components of HBM.

3. The inclusion criteria states that participants must be between 15-35. Can the authors provide a rationale for this large age gap? This is a wide range and the health behaviors of adults in their 20s/30s are likely to be different from individuals in their teens.

Response: As it is described in line 127-128 of the method section, the inclusion criteria is 18 to 35 not 15 to 35. We concur with your idea that there might still be variations. Nevertheless, in the Ethiopian educational system those who completed grade 10 are allowed to join technical and vocational colleges if they couldn’t achieve sufficient scores to join preparatory schools (grade 11 and 12). On the other hand adults also can join these technical and vocational colleges. Therefore, the age pattern in the colleges is mixed. Most probably, students in the same college, school or university behave in the same manner. However, as you said it well, we also expect variation in the health behavior. Thus, instead of excluding those aged 18 to 19 from the study, we preferred to include them and perform the subgroup analysis and compare the results. During the analysis, we will make this age group in one category.

4. The authors did include a sample message, but I would like to see more examples of the content that will be used in this study. Given that this is a messaging study, there needs to be more explanation of the actual messages and what they look like.

Response: Thank you again for your comment. We added more examples of text messages in the intervention section from line number 175 to 188 which include each behavioral risk factor of cancer and the components of HBM.

---

## [Decision Letter · Decision Letter 2]

1 Feb 2021

PONE-D-20-05561R2

The effect of mobile text messages on behavioral risk factors of cancer among college students, Northeast Ethiopia: a randomized controlled trial protocol

PLOS ONE

Dear Dr. Hussien,

Thank you for submitting your manuscript to PLOS ONE. After careful consideration, we feel that it has merit but does not fully meet PLOS ONE’s publication criteria as it currently stands. Therefore, we invite you to submit a revised version of the manuscript that addresses the points raised during the review process.

Please respond to the comments from Reviewer 3. Specifically, please copyedit the manuscript paying special attention to the messages being tested. Please carefully consider the reviewer's advice on the message content.

We look forward to receiving your revised manuscript.

Kind regards,

Nancy Beam, PhD

Staff Editor

PLOS ONE

Reviewers' comments:

Reviewer's Responses to Questions

**Comments to the Author**

1. Does the manuscript provide a valid rationale for the proposed study, with clearly identified and justified research questions?

Reviewer #3: Yes

2. Is the protocol technically sound and planned in a manner that will lead to a meaningful outcome and allow testing the stated hypotheses?

Reviewer #3: Yes

3. Is the methodology feasible and described in sufficient detail to allow the work to be replicable?

Reviewer #3: Yes

4. Have the authors described where all data underlying the findings will be made available when the study is complete?

Reviewer #3: Yes

5. Is the manuscript presented in an intelligible fashion and written in standard English?

Reviewer #3: Yes

6. Review Comments to the Author

You may also provide optional suggestions and comments to authors that they might find helpful in planning their study.

Reviewer #3: I think this manuscript pending two criteria are met:

The authors proofread the added content about the HBM within lines 75-83 since there are a few grammatical errors. For example, "believe" should be "belief." The authors should also include at least one or two lines mentioning self-efficacy and cues to action given that these are variables within the HBM, but are not mentioned here.

The authors should carefully proofread the messages that they plan to use, which are listed within lines 175-188. This is critical given that this is a messaging study. Additionally, although this is perhaps beyond the scope of this study, I recommend that the authors be careful about using messages like, "cancer is fatal had no certain cure" since this fear-based and might lead to the recipient tuning out the message completely. Also, the message, "although you are addicted to drink alcohol, you can decrease the amount and quit it through time” is not entirely accurate (and also needs to be proofread) since people who are addicted to alcohol cannot always decrease their consumption and/or quit over time. Finally, messages like, "avoid laziness!" might be seen as condescending and can produce a boomerang effect. Some of these messages also fail to consider structural/social determinants of health and put all of the emphasis on individual behavior change. Again, I'm not sure if all of this can be addressed here, but these are points that should be kept in mind while moving forward with this project.

The authors have addressed my other concerns and I do not need to review another version of this (although I am happy to do so if the editor prefers) because I trust that the authors will make these changes.

7. PLOS authors have the option to publish the peer review history of their article (what does this mean?). If published, this will include your full peer review and any attached files.

Reviewer #3: No

---

## [Author Response · Author response to Decision Letter 2]

10 Feb 2021

Response to Reviewers

PONE-D-20-05561R2

The effect of mobile text messages on behavioral risk factors of cancer among college students, northeast Ethiopia: a randomized controlled trial protocol

PLOS ONE

On behalf of all authors, I would like to thank editors and reviewers for their continued constructive comments and suggestions to improve the excellence of this manuscript. We revised the manuscript based on the comments. We hope that we addressed all the comments and questions. Here is a point by point response for each comment and question.

Reviewer #3:

1. The authors proofread the added content about the HBM within lines 75-83 since there are a few grammatical errors. For example, "believe" should be "belief."

Response: Thank you for your suggestion. We revised the suggested content and the whole manuscript for grammar and spelling. 

2. The authors should also include at least one or two lines mentioning self-efficacy and cues to action given that these are variables within the HBM, but are not mentioned here.

Response: Thank you for your suggestion. In the introduction section lines 85 and 86 stated about self-efficacy. We also added more descriptions from lines 81 to 83 which explains self-efficacy and cues to action.

3. The authors should carefully proofread the messages that they plan to use, which are listed within lines 175-188. This is critical given that this is a messaging study. Additionally, although this is perhaps beyond the scope of this study, I recommend that the authors be careful about using messages like, "cancer is fatal had no certain cure" since this fear-based and might lead to the recipient tuning out the message completely. Also, the message, "although you are addicted to drink alcohol, you can decrease the amount and quit it through time” is not entirely accurate (and also needs to be proofread) since people who are addicted to alcohol cannot always decrease their consumption and/or quit over time. Finally, messages like, "avoid laziness!" might be seen as condescending and can produce a boomerang effect. Some of these messages also fail to consider structural/social determinants of health and put all of the emphasis on individual behavior change. 

Response: Thank you for your suggestion. In fact, the messages will be translated and revised critically by experts before sending to the participants. We modified the sample messages as suggested.

---

## [Decision Letter · Decision Letter 3]

24 Mar 2021

PONE-D-20-05561R3

The effect of mobile text messages on behavioral risk factors of cancer among college students, Northeast Ethiopia: a randomized controlled trial protocol

PLOS ONE

Dear Dr. Hussien,

Thank you for submitting your manuscript to PLOS ONE. Apologies for the delay in returning this to you. The orignial editor was unavailable so this paper was passed on to myself for review, and we struggled to identify reviewsers. After careful consideration, we feel that it has merit but does not fully meet PLOS ONE’s publication criteria as it currently stands. Therefore, we invite you to submit a revised version of the manuscript that addresses the points raised during the review process.

This is a minor but important revision: given reviewer comments, it is imperative to highlight likely problems and biases as identified by reviewer comments more clearly in your disucssion (and/or justifiy better the reasons for the chosen study design, intervention development and operationalisation in the methods). This should also help the framing of the write-up of your completed study.

We look forward to receiving your revised manuscript.

Kind regards,

Lion Shahab, MA MSc MSc PhD CPsychol

Academic Editor

PLOS ONE

Journal Requirements:

Reviewers' comments:

Reviewer's Responses to Questions

**Comments to the Author**

1. Does the manuscript provide a valid rationale for the proposed study, with clearly identified and justified research questions?

Reviewer #3: Yes

Reviewer #4: Yes

Reviewer #5: Yes

2. Is the protocol technically sound and planned in a manner that will lead to a meaningful outcome and allow testing the stated hypotheses?

Reviewer #3: Yes

Reviewer #4: Partly

Reviewer #5: Yes

3. Is the methodology feasible and described in sufficient detail to allow the work to be replicable?

Reviewer #3: Yes

Reviewer #4: No

Reviewer #5: Yes

4. Have the authors described where all data underlying the findings will be made available when the study is complete?

Reviewer #3: Yes

Reviewer #4: Yes

Reviewer #5: No

5. Is the manuscript presented in an intelligible fashion and written in standard English?

Reviewer #3: Yes

Reviewer #4: Yes

Reviewer #5: Yes

6. Review Comments to the Author

You may also provide optional suggestions and comments to authors that they might find helpful in planning their study.

Reviewer #3: I appreciate the authors' efforts to revise the manuscript and for their attention to my comments. I "accept" the manuscript in its current form and the authors have addressed my previous concerns. However, I want to reiterate the importance of having the messages reviewed by communication/messaging experts who have experience with text-messaging studies aimed at health behavior change, so that they can offer suggestions for framing, wording, etc. This study has the potential to make a unique and important contribution to the text-messaging/mobile health literature. I look forward to reading the findings once this is implemented and the data is analyzed.

Reviewer #4: Thank you for the opportunity to review this protocol, which outlines the study procedures and analysis plan for an RCT of a text message intervention informed by the Health Belief Model to increase the awareness of behavioural risk factors for cancer in college students in Ethiopia. The protocol has several strengths, including the development of the intervention on the basis of health psychology theory and a sound statistical analysis plan. However, I have highlighted some concerns below for the authors’ consideration.

Major comments

1. As students in the same college are individually randomised to the intervention and control arms, there’s a large risk of contamination (e.g. students may simply compare the text messages received). Due to the target population and setting of interest, this study would have benefitted from the use of a cluster RCT design. However, as recruitment has already been completed, this should at the very least be discussed as an important limitation of the study design. Linked to this, as recruitment is already completed, I don’t see how this can usefully be published as a Registered Report Protocol, as there’s insufficient scope to improve on the study design.

2. When setting up the rationale for the study in the Introduction, a better lay of the land is needed. It would be useful to know the proportion of adults and/or students in Ethiopia with access to mobile phones. There’s a vast literature on the effectiveness of text messages for smoking cessation, physical activity, etc. (please see a suggested reference to a review of systematic reviews below). It would be important to refer to this wide literature in the Introduction, that positive effects on behaviour have been observed, and then state that few existing interventions have been evaluated in LMICs, including Ethiopia (off the top of my head, I’m only aware of a couple for smoking cessation, with one in China and one in Vietnam):

https://www.ncbi.nlm.nih.gov/pmc/articles/PMC4406229/

https://pubmed.ncbi.nlm.nih.gov/26730560/

https://www.ncbi.nlm.nih.gov/pmc/articles/PMC7381024/

3. Although it’s a compelling aspect of the study that the intervention has been developed on the basis of the Health Belief Model, it’s unclear how the intervention messages map onto the different components of the HBM. It would be useful to make this more explicit, for example by including a table with example messages and indications in a second column as to how each message maps onto the different components of the model. At present, it’s unclear whether all components were covered by the messages. This is important to understand, as we wouldn’t expect the intervention to have an effect on, for example, self-efficacy, if the intervention messages themselves don’t directly attempt to boost self-efficacy.

4. It’s unclear why participants aren’t recruited on the basis of engaging in one or multiple unhealthy behaviours? There’s arguably no need to raise awareness that excessive alcohol consumption is a risk factor for cancer in those who don’t engage in heavy drinking. It’s also unclear why the primary outcome isn’t behaviour change – this would be of greater interest to the public health community and a more typical outcome for an RCT.

Specific comments

Introduction

5. The second sentence about cancer incidence and mortality refers to global cancer statistics but the citation refers to cancer rates in Ethiopia. Please change to a more suitable reference, e.g. https://acsjournals.onlinelibrary.wiley.com/doi/full/10.3322/caac.21660

6. It would be useful to more explicitly state that although there are many different risk factors for cancer (e.g. behavioural, biological, environmental), not all of these are modifiable, and that this study focuses specifically on modifiable health behaviours. Environmental exposure to, for example, air pollutants, and soil and drinking water contaminants, could theoretically also be addressed via behaviour change (although may require wider policy changes to impact behaviour).

7. Line 74 – there are some typos here, e.g. “The” is missing at the beginning of the sentence; “health related” rather than “healthy related”.

8. Line 85 – “and by their confidence that the recommended action” – this is repeating information on line 79; please remove.

9. Line 86 – “Thus, we will use the HBM to measure the perception…” – this doesn’t fully elaborate on the study aims, which also includes the examination of behaviour change. Given this is described in the next paragraph, I’d suggest changing this to a broad statement that the HBM will therefore be used to develop an intervention to increase awareness of and perceived susceptibility to cancer, and affect change to four key health behaviours.

10. Rather than stating both hypotheses and objectives for the study, I’d suggest including a list of the key research questions and how they relate to the different follow-up assessments and outcome variables of interest. For example, the first research question may be whether participants allocated to the intervention arm, compared with those allocated to the control arm, have increased knowledge of the cancer risks of smoking/alcohol/etc at a 6-month follow-up. Please list all of the main research questions and end points.

Methods

11. Although a CONSORT flow chart is included, it would be useful to reference the CONSORT checklist in the ‘Study design’ section and mention that this was used in the design and reporting of the study: http://www.consort-statement.org/

12. The study setting section describes the number of staff members but the eligibility criteria state that only students are eligible to take part – it’s unclear why staff members aren’t also eligible? Also, it’s unclear why only those aged 18-35 years are eligible to take part? Please provide a rationale and remove any superfluous information about the employees in the ‘Study setting’ section (as they aren’t part of the target population).

13. The authors state that the study is powered to detect a difference in “knowledge and perception of cancer risk” but then go on to state that the power calculation is based on perceived self-efficacy – these are not identical psychological constructs, so it would be important to clarify which of these is the primary outcome for the study. It would also be useful to contextualise the 2.9-point expected difference in knowledge or self-efficacy – is this on a 5-point or 7-point Likert scale?

14. It’s unclear what this means “Sample size will be allocated to…” – does this mean only two colleges will be used for recruitment? If so, this needs to match the above ‘Study setting’ section.

15. “Students’ IDs will be obtained…” – do you mean that all students with an ID will be contacted and asked if they are willing to join the trial?

16. “Then; simple random sampling method will be used to select eligible study subjects” – this sounds incorrect. Surely interested students will be assessed for eligibility via a screening survey, with eligible participants randomised with simple randomisation to the intervention or control arm?

17. Blinding – it should also be stated that the researchers will be blinded to the allocation sequence (i.e. single blind).

18. Control – please provide a bit more information on what is considered a ‘general health message’ in this context. I assume these won’t mention the four health behaviours of interest?

19. Primary outcomes – please comment on whether the 5-point Likert scale has been validated.

20. Secondary outcomes – please provide references to the specific food frequency questionnaire that will be used and whether it has been validated, the IPAQ-S and whether it has been validated, the NIAAA tool to assess alcohol consumption and the WHO tool to assess smoking. I’d recommend using the AUDIT or AUDIT-C for alcohol: https://apps.who.int/iris/handle/10665/67205 It’s unclear whether the smoking assessment will examine cigarette smoking only or any kind of tobacco?

21. The proposed statistical analysis plan looks good.

Reviewer #5: Dear PLOS ONE Editors,

Thank you for the opportunity to review an article entitled “The effect of mobile text messages on behavioral risk factors of cancer among college students, Northeast Ethiopia: a randomized controlled trial protocol” prepared by Hussien and colleagues.

This manuscript has already undergone peer review. It is well written and provides the relevant information for a trial protocol. However, I have a number of suggestions for the authors to consider for how to improve the proposed intervention and the trial, if still possible.

[1] The authors selected Health Belief Model (HBM) as a theoretical basis for the intervention. The constructs specified by HBM can contribute to risky behaviours and behaviour change (e.g. beliefs about what are risk factors), but they primarily focus motivational factors. However, there are other important determinant of behaviour change – as mentioned by other reviewers earlier (e.g. social determinants). For example, another model of behaviour – Capability, Opportunity, and Motivation (COM-B) shows that for any behaviour change to take place one not only needs motivation, but also capability (knowledge, skills) as well as opportunity (social and physical). Therefore, an intervention that only target some of the COM-B elements without addressing other known barriers is likely not to be sufficiently effective. The evaluation survey is accounting for it (e.g. by asking about income sufficient to change behaviour), but it does not seem that the intervention suggest ways of changing behaviour when one has insufficient income. Can the authors still consider and address the limitations of HBM and if the intervention content could still be improved?

[2] I see some further potentially missed opportunities for increasing the effectiveness of the intervention. For example, important and effective components (or behaviour change techniques) of behaviour change interventions are setting goals, monitoring and feedback. Without specifying goals for behaviour change early on for the participants it will be quite improbable that they will engage with the intervention and attempt behaviour change (the attempts being a first step to change the behaviour).

[3] The intervention targets are very broad (and address a number of health behaviours) – ideally the content of the interventions would be tailored to the risk behaviours of that individuals to increase the relevance of the intervention. Previous SMS texting interventions that target a single behaviour (e.g. smoking cessation) were very complex and detailed, and it is unlikely that only a few sms text about one behaviour will be sufficient to change that behaviour.

[4] Students in public colleges are more educated and technology-savvy group in the population. Additionally, they may also be at lower risk of engaging in some of the risky behaviours, and given their young age their cancer risk is also much lower. It would be important for the authors to clarify the benefit of testing this intervention among this population, rather than among adults in the general population for whom it may be more relevant and who may benefit more from it.

[5] Additionally, the eligibility criteria do not specify that the participants should engage in any unhealthy behaviour. It is therefore possible that all of the participants will lead a healthy lifestyle and have no unhealthy behaviours. This is another important limitation of the study that will make it more difficult to show any effect of the proposed intervention. Can the authors consider to first screen participants for unhealthy behaviours and recruit only those who exhibit behaviours and the levels of these behaviours (e.g. smoking, high alcohol consumption, etc) that can realistically show any reduction or improvement during the trial?

[6] Texting has been shown to be a cost-effective medium to deliver education or more complex behaviour change interventions for a number of health risk behaviours. Texting as a technology can have more reach than newer technologies (e.g. smartphones app) as they can be delivered on any mobile device. However, sms texts can be ignored, the options for content are very limited, and there is little or no possibility to assess engagement with the texts objectively. I encourage the authors to include more discussion around the pros and cons of using texting, and how some of the limitations for the interventions and evaluation will be addressed.

If the authors are no longer able to change the intervention or the study design, then it would be very helpful for the published trial protocol to include a discussion of these important limitations and highlight the contribution that this study still brings despite the known limitations.

With best wishes,

Dr Aleksandra Herbec

7. PLOS authors have the option to publish the peer review history of their article (what does this mean?). If published, this will include your full peer review and any attached files.

Reviewer #3: No

Reviewer #4: No

Reviewer #5: **Yes: **Dr Aleksandra Herbec

---

## [Author Response · Author response to Decision Letter 3]

26 Apr 2021

Response to Reviewers

PONE-D-20-05561R3 

The effect of mobile text messages on behavioral risk factors of cancer among college students, Northeast Ethiopia: a randomized controlled trial protocol

PLOS ONE

First of all, on behalf of all authors, I wish to thank the editor and the reviewers for their valuable and constructive comments to improve the excellence of the manuscript. The manuscript went through a long review process since February 2020 and was reviewed by three reviewers. We revised it thoroughly and addressed all issues raised by reviewers. You can refer all the versions of the manuscript and response to reviewers. Due to the delay from the review process, we couldn’t start the intervention yet. Now, we are under pressure from Wollo University, the institution which partially funded the project, to start implementation and finalize the output. It’s taking unacceptably long time for just the protocol. We also consider the issues raised in the current review very important and we revised the manuscript based on the comments and issues raised by the editor and the reviewers for one last time. We are confident that we have addressed all these comments adequately and we therefore hope that the manuscript is accepted for publication. We kindly request the final decision for the manuscript. Find hereunder a point by point reply to the comments and questions raised by the editor and all reviewers. 

Editor

Response: Thank you for your suggestions. We have reviewed the manuscript particularly the reference list to meet the journal requirements and we hope that the manuscript in its current form does so. We did not use any papers that are retracted.

Reviewer # 3

1. I appreciate the authors' efforts to revise the manuscript and for their attention to my comments. I "accept" the manuscript in its current form and the authors have addressed my previous concerns. However, I want to reiterate the importance of having the messages reviewed by communication/messaging experts who have experience with text-messaging studies aimed at health behavior change, so that they can offer suggestions for framing, wording, etc. This study has the potential to make a unique and important contribution to the text-messaging/mobile health literature. I look forward to reading the findings once this is implemented and the data is analyzed.

Response:

Thank you for your constructive comments throughout the review process. The manuscript improved a lot from your feedback. As you suggested, once the protocol is accepted, we will start developing intervention tools (messages) in consultation with experts in communication and health education.

Reviewer # 4

1. As students in the same college are individually randomized to the intervention and control arms, there’s a large risk of contamination (e.g. students may simply compare the text messages received). Due to the target population and setting of interest, this study would have benefitted from the use of a cluster RCT design. However, as recruitment has already been completed, this should at the very least be discussed as an important limitation of the study design. Linked to this, as recruitment is already completed, I don’t see how this can usefully be published as a Registered Report Protocol, as there’s insufficient scope to improve on the study design.

Response:

Thank you for your comment. We agree that there could be intervention contamination in the trial. We have been discussing this issue within the team. However, due to the huge variation in the nature of colleges (technical vs teachers’ education college), cluster randomization will lead to selection bias. As a result, we decided to use individual randomization and instruct individual participants on the trial process thoroughly. We will discuss further the limitations of the study during the result report. We conducted a preliminary assessment and are now waiting for the protocol to be accepted before starting implementation of the intervention.

2. When setting up the rationale for the study in the Introduction, a better lay of the land is needed. It would be useful to know the proportion of adults and/or students in Ethiopia with access to mobile phones. There’s a vast literature on the effectiveness of text messages for smoking cessation, physical activity, etc. (please see a suggested reference to a review of systematic reviews below). It would be important to refer to this wide literature in the Introduction, that positive effects on behaviour have been observed, and then state that few existing interventions have been evaluated in LMICs, including Ethiopia (off the top of my head, I’m only aware of a couple for smoking cessation, with one in China and one in Vietnam):

https://www.ncbi.nlm.nih.gov/pmc/articles/PMC4406229/

https://pubmed.ncbi.nlm.nih.gov/26730560/

https://www.ncbi.nlm.nih.gov/pmc/articles/PMC7381024/

Response:

We do not have data on the coverage of mobile phones for the general population in Ethiopia. Nevertheless, our preliminary assessment indicated that almost all college students have access for it. Thank you for your suggestion of articles on the effectiveness of mobile text message on smoking cessation and health. We incorporated some of the articles in lines 97 to 99 of the introduction section. However, as far as we are aware, there is no study that evaluated the effectiveness in Ethiopia. We are very much grateful if you provide us with the link. 

3. Although it’s a compelling aspect of the study that the intervention has been developed on the basis of the Health Belief Model, it’s unclear how the intervention messages map onto the different components of the HBM. It would be useful to make this more explicit, for example by including a table with example messages and indications in a second column as to how each message maps onto the different components of the model. At present, it’s unclear whether all components were covered by the messages. This is important to understand, as we wouldn’t expect the intervention to have an effect on, for example, self-efficacy, if the intervention messages themselves don’t directly attempt to boost self-efficacy.

Response:

Thank you for your suggestion. We didn’t start the intervention yet. Once the protocol is accepted, we will proceed developing messages in collaboration with experts in health education. As this is a protocol, we provided sufficient sample messages for each component of HBM in table one of the manuscript. 

4. It’s unclear why participants aren’t recruited on the basis of engaging in one or multiple unhealthy behaviors? There’s arguably no need to raise awareness that excessive alcohol consumption is a risk factor for cancer in those who don’t engage in heavy drinking. It’s also unclear why the primary outcome isn’t behavior change – this would be of greater interest to the public health community and a more typical outcome for an RCT.

Response:

People might not be involved in heavy alcohol drinking or smoking at the moment. However, it does not necessarily mean that they have good knowledge on the risk of alcohol consumption and smoking on cancer. If people are not involved in unhealthy lifestyles, having good knowledge would help them to continue with such a healthy lifestyle. Those who don’t have good knowledge on the risk of physical inactivity, unhealthy dietary habits, smoking, and excessive alcohol consumption might be involved in such lifestyle sometime later. Therefore, our target group is not necessarily those who are practicing an unhealthy lifestyle. Nevertheless, we have planned to recruit more samples and take those with knowledge scores below two-third will be randomized. Our primary outcome is knowledge on cancer and lifestyle risk factors. We think that the reviewer did not realize the context of Ethiopia that people are not aware of behavioral risks of cancer. A significant proportion of the population still consider cancer as a disease related with bad spirits. Hence, it’s crucial to improve the knowledge before the actual behavior. Moreover, as our project is for 6 months duration, the effect on the change in behavior might not be large. In general, we will see the effect of our intervention on knowledge as primary outcome and assess behavior on four major risks as secondary outcome. Based on our result, we will consider the change in behavior in the scale up project for a longer duration. 

Specific comments

Introduction

5. The second sentence about cancer incidence and mortality refers to global cancer statistics but the citation refers to cancer rates in Ethiopia. Please change to a more suitable reference, e.g. 

https://acsjournals.onlinelibrary.wiley.com/doi/full/10.3322/caac.21660

Response:

Thank you for your suggestion. We revised accordingly. 

6. It would be useful to more explicitly state that although there are many different risk factors for cancer (e.g. behavioral, biological, environmental), not all of these are modifiable, and that this study focuses specifically on modifiable health behaviours. Environmental exposure to, for example, air pollutants, and soil and drinking water contaminants, could theoretically also be addressed via behaviour change (although may require wider policy changes to impact behaviour).

Response:

We agree that environmental factors are also among the potentially modifiable risk factors. We revised it on page 68 to 69 of the revised manuscript. However, our main interest in this intervention is to improve knowledge, perception and behavior (to some extent) towards lifestyle risks of cancer. It is not possible to address every aspect of cancer risk factors with this short period and limited funding. Probably, we will consider to look at these issues in the follow up project.

7. Line 74 – there are some typos here, e.g. “The” is missing at the beginning of the sentence; “health related” rather than “healthy related”.

Response:

Thank you for your suggestion. We revised accordingly. 

8. Line 85 – “and by their confidence that the recommended action” – this is repeating information on line 79; please remove.

Response:

Thank you for your suggestion. We revised it accordingly. 

9. Line 86 – “Thus, we will use the HBM to measure the perception…” – this doesn’t fully elaborate on the study aims, which also includes the examination of behaviour change. Given this is described in the next paragraph, I’d suggest changing this to a broad statement that the HBM will therefore be used to develop an intervention to increase awareness of and perceived susceptibility to cancer, and affect change to four key health behaviors.

Response:

Thank you for your suggestion and we revised it accordingly.

10. Rather than stating both hypotheses and objectives for the study, I’d suggest including a list of the key research questions and how they relate to the different follow-up assessments and outcome variables of interest. For example, the first research question may be whether participants allocated to the intervention arm, compared with those allocated to the control arm, have increased knowledge of the cancer risks of smoking/alcohol/etc at a 6-month follow-up. Please list all of the main research questions and end points.

Response:

As suggested, we added the main research questions. However, the primary and secondary end points are sufficiently described in detail in lines 199 to 227 of the method section. We think that putting the end points again is no more than a repetition of primary and secondary outcomes.

Methods

11. Although a CONSORT flow chart is included, it would be useful to reference the CONSORT checklist in the ‘Study design’ section and mention that this was used in the design and reporting of the study: http://www.consort-statement.org/

Response:

We cited the reference and thanks for the suggestion.

12. The study setting section describes the number of staff members but the eligibility criteria state that only students are eligible to take part – it’s unclear why staff members aren’t also eligible? Also, it’s unclear why only those aged 18-35 years are eligible to take part? Please provide a rationale and remove any superfluous information about the employees in the ‘Study setting’ section (as they aren’t part of the target population).

Response:

Our aim is to see the effectiveness on students, which are more or less homogeneous groups. The staff and students are somehow different in terms of lifestyle and socioeconomic characteristics, which is not our interest to observe in this study. Hence, we removed the information about the number of staff as suggested.

13. The authors state that the study is powered to detect a difference in “knowledge and perception of cancer risk” but then go on to state that the power calculation is based on perceived self-efficacy – these are not identical psychological constructs, so it would be important to clarify which of these is the primary outcome for the study. It would also be useful to contextualise the 2.9-point expected difference in knowledge or self-efficacy – is this on a 5-point or 7-point Likert scale?

Response:

We didn’t calculate power, we just determined (assumed) 80% power to detect a 2.9 mean difference. Nevertheless, we corrected the editing problem that we calculated the sample size considering the self-efficacy. We will use 5-point Likert scale. 

14. It’s unclear what this means “Sample size will be allocated to…” – does this mean only two colleges will be used for recruitment? If so, this needs to match the above ‘Study setting’ section.

Response:

There are three colleges in the town. One of them is Health Science College, which will be excluded as students from Health Science College are expected to have a different (higher) knowledge related to cancer and risks. Thus, the sample size will be allocated to the two remaining colleges (Woizero Siheen Polytechnic College and Dessie College of teacher’s education). 

15. “Students’ IDs will be obtained…” – do you mean that all students with an ID will be contacted and asked if they are willing to join the trial?

Response:

As we clearly described, the students’ ID number will be used for random sampling (line 152 to 154). The confusion may be using the term ‘ID’ instead of ‘ID number’ and we modified it. 

“Students’ ID number will be obtained from the registrar's office of each college and used as a sampling frame. Then; a simple random sampling method will be used to select study subjects.”

16. “Then; simple random sampling method will be used to select eligible study subjects” – this sounds incorrect. Surely interested students will be assessed for eligibility via a screening survey, with eligible participants randomized with simple randomisation to the intervention or control arm?

Response:

The simple random sampling here is not ‘the randomization to intervention and control arm’. Rather we are referring to the random sampling to be included in the study since we won’t enroll all students. From all eligible participants, we will select random samples for the study. Then, we will randomize either to the intervention or control arm using simple randomization (see lines 160 to 163 of the recruitment and randomization section). We removed the word ‘eligible’ to avoid confusion.

17. Blinding – it should also be stated that the researchers will be blinded to the allocation sequence (i.e. single blind).

Response:

We will use single blinding, in which only the study participants will be blinded. In such kind of interventions, it’s not feasible for investigators to be blinded. If investigators are also blinding, it is double blinding. Furthermore, the most important thing is not using the term ‘single’ or ‘double’ rather the description of who will be blinded (https://www.bmj.com/rapid-response/2011/10/31/what-single-blind-trial). We clearly mentioned in line 176 to 178 that study participants will be blinded to the intervention arm.

18. Control – please provide a bit more information on what is considered a ‘general health message’ in this context. I assume these won’t mention the four health behaviours of interest?

Response:

General health messages are contents other than the four mentioned behaviors and not related to cancer and the four lifestyle risks. The focus for the general health messages will be mainly related to infectious diseases such as sanitation, how to prevent HIV and TB, etc. 

19. Primary outcomes – please comment on whether the 5-point Likert scale has been validated.

Response: it is validated in the article https://www.ncbi.nlm.nih.gov/pmc/articles/PMC5454653/

20. Secondary outcomes – please provide references to the specific food frequency questionnaire that will be used and whether it has been validated, the IPAQ-S and whether it has been validated, the NIAAA tool to assess alcohol consumption and the WHO tool to assess smoking. I’d recommend using the AUDIT or AUDIT-C for alcohol: https://apps.who.int/iris/handle/10665/67205 It’s unclear whether the smoking assessment will examine cigarette smoking only or any kind of tobacco?

Response: 

The IPAQ short is internationally validated in different context including low income countries.

For tobacco use, we will consider any kind of tobacco.

Thank you for your suggestion. We will use AUDIT to assess alcohol consumption.

We revised and cited each tool accordingly. 

21. The proposed statistical analysis plan looks good.

Response:

Thank you.

Reviewer #5

Dear PLOS ONE Editors,

Thank you for the opportunity to review an article entitled “The effect of mobile text messages on behavioral risk factors of cancer among college students, Northeast Ethiopia: a randomized controlled trial protocol” prepared by Hussien and colleagues.

This manuscript has already undergone peer review. It is well written and provides the relevant information for a trial protocol. However, I have a number of suggestions for the authors to consider for how to improve the proposed intervention and the trial, if still possible.

1. The authors selected Health Belief Model (HBM) as a theoretical basis for the intervention. The constructs specified by HBM can contribute to risky behaviours and behaviour change (e.g. beliefs about what are risk factors), but they primarily focus motivational factors. However, there are other important determinant of behaviour change – as mentioned by other reviewers earlier (e.g. social determinants). For example, another model of behaviour – Capability, Opportunity, and Motivation (COM-B) shows that for any behaviour change to take place one not only needs motivation, but also capability (knowledge, skills) as well as opportunity (social and physical). Therefore, an intervention that only target some of the COM-B elements without addressing other known barriers is likely not to be sufficiently effective. The evaluation survey is accounting for it (e.g. by asking about income sufficient to change behaviour), but it does not seem that the intervention suggest ways of changing behaviour when one has insufficient income. Can the authors still consider and address the limitations of HBM and if the intervention content could still be improved?

Response:

Thank you for the suggestion. As we described in lines 201 to 203, socioeconomic characteristics will be assessed and be considered to adjust using GEE in the analysis section. Regarding (COM-B), it is not our aim in this study to look at it. The COM-B model of behavior is to identify what needs to change in order for a behavior change intervention to be effective. Our aim is not on the process, rather on the outcome. The main reason to conduct a randomized study is to control for some of the important confounding variables. Once the most important variables are being adjusted, the rest residual confounders will be expected to be controlled in the randomization. Furthermore, the change in behavior is the secondary outcome, the primary end point of this study is change in cancer related knowledge.

2. I see some further potentially missed opportunities for increasing the effectiveness of the intervention. For example, important and effective components (or behaviour change techniques) of behaviour change interventions are setting goals, monitoring and feedback. Without specifying goals for behaviour change early on for the participants it will be quite improbable that they will engage with the intervention and attempt behaviour change (the attempts being a first step to change the behaviour).

Response:

We agree that components including setting goals, monitoring and feedback are crucial in behavioral change interventions. However, incorporating such components is not feasible in this short term study using mobile text messages. We cannot address everything in a single short term study. Our main aim is whether sending cancer specific text messages improve the knowledge (primary), perception and practice (secondary ) towards behavioral risks of cancer. Once again, our primary outcome is not a change in behavior. We will observe the change in behavior as a secondary outcome. 

3. The intervention targets are very broad (and address a number of health behaviours) – ideally the content of the interventions would be tailored to the risk behaviours of that individuals to increase the relevance of the intervention. Previous SMS texting interventions that target a single behaviour (e.g. smoking cessation) were very complex and detailed, and it is unlikely that only a few sms text about one behaviour will be sufficient to change that behaviour.

Response:

We agree that a change in behavior needs a more comprehensive and multi-component intervention considering various aspects of complex human behavior. However, our main outcome is change in knowledge and perception. We will evaluate the change in behavior as a secondary outcome. We clearly elaborated the primary and secondary endpoints in page 8 and 9 of the method section.

4. Students in public colleges are more educated and technology-savvy group in the population. Additionally, they may also be at lower risk of engaging in some of the risky behaviours, and given their young age their cancer risk is also much lower. It would be important for the authors to clarify the benefit of testing this intervention among this population, rather than among adults in the general population for whom it may be more relevant and who may benefit more from it.

Response:

The main reason is, for a text message intervention participants should be able to read and write and have basics of mobile use. Hence, we will assess the feasibility and practicality to scale up this intervention to the general population in the future if it is effective among college students. 

5. Additionally, the eligibility criteria do not specify that the participants should engage in any unhealthy behaviour. It is therefore possible that all of the participants will lead a healthy lifestyle and have no unhealthy behaviours. This is another important limitation of the study that will make it more difficult to show any effect of the proposed intervention. Can the authors consider to first screen participants for unhealthy behaviours and recruit only those who exhibit behaviours and the levels of these behaviours (e.g. smoking, high alcohol consumption, etc) that can realistically show any reduction or improvement during the trial?

Response:

We think that there is mis-understanding of our primary aim. Our primary outcome is to see the change in cancer related knowledge and perception. We will observe the change in behavior as a secondary outcome. From our previous experience, it is unlikely that all of the participants have no unhealthy behaviors. Furthermore, not being involved in unhealthy behavior does not necessarily mean that they have good knowledge on cancer. Having good knowledge would help those with unhealthy lifestyles to change and those with a healthy lifestyle to persist in that behavior. Hence, our target groups are all college students, not specifically those involved in unhealthy lifestyles.

6. Texting has been shown to be a cost-effective medium to deliver education or more complex behaviour change interventions for a number of health risk behaviours. Texting as a technology can have more reach than newer technologies (e.g. smartphones app) as they can be delivered on any mobile device. However, sms texts can be ignored, the options for content are very limited, and there is little or no possibility to assess engagement with the texts objectively. I encourage the authors to include more discussion around the pros and cons of using texting, and how some of the limitations for the interventions and evaluation will be addressed.

If the authors are no longer able to change the intervention or the study design, then it would be very helpful for the published trial protocol to include a discussion of these important limitations and highlight the contribution that this study still brings despite the known limitations.

Response:

Thank you for your suggestion. We added as suggested from line 307 to 310. Moreover, whether the participants looked at the texts and how frequent they checked for their texts will be assessed post-intervention. Thus participants’ engagement will be assessed later and be discussed in the limitation of the study results.

---

## [Decision Letter · Decision Letter 4]

7 Jun 2021

PONE-D-20-05561R4

The effect of mobile text messages on knowledge and perception towards cancer and behavioral risks among college students, Northeast Ethiopia: a randomized controlled trial protocol

PLOS ONE

Dear Dr. Hussien,

Thank you for submitting your manuscript to PLOS ONE. After careful consideration, we feel that it has merit but does not fully meet PLOS ONE’s publication criteria as it currently stands. Therefore, we invite you to submit a revised version of the manuscript that addresses the points raised during the review process.

As you will see from the reviewer comments below, only very minor further revisions are requested, namely 1) to change minor inaccuracies in the manuscript and 2) to add some of the justifications provided in the responses to reviewers regarding decisions on study design in the actual manuscript.

We look forward to receiving your revised manuscript.

Kind regards,

Lion Shahab, MA MSc MSc PhD CPsychol

Academic Editor

PLOS ONE

Journal Requirements:

Reviewers' comments:

Reviewer's Responses to Questions

**Comments to the Author**

1. Does the manuscript provide a valid rationale for the proposed study, with clearly identified and justified research questions?

Reviewer #4: Yes

Reviewer #5: Yes

2. Is the protocol technically sound and planned in a manner that will lead to a meaningful outcome and allow testing the stated hypotheses?

Reviewer #4: Yes

Reviewer #5: Yes

3. Is the methodology feasible and described in sufficient detail to allow the work to be replicable?

Reviewer #4: Yes

Reviewer #5: No

4. Have the authors described where all data underlying the findings will be made available when the study is complete?

Reviewer #4: Yes

Reviewer #5: No

5. Is the manuscript presented in an intelligible fashion and written in standard English?

Reviewer #4: Yes

Reviewer #5: Yes

6. Review Comments to the Author

You may also provide optional suggestions and comments to authors that they might find helpful in planning their study.

Reviewer #4: The authors have done a fine job addressing the reviewer comments. I only have two remaining points. In the abstract, "To test the change in score of HBM" is unclear and should be changed to "To test the change in knowledge and perception" or similar. The 'Trial status' section still says that "We are currently recruiting and enrolling students in the study. As of February 2020, we have enrolled 160 participants." Given the authors' response that the intervention is yet to be developed and delivered, please amend this accordingly.

Reviewer #5: I would like to thank the Authors for responding in detail to my suggestions and questions. It is very helpful to see it clarified that the primary outcome are changes in knowledge/perceptions, rather than behaviours. I accept most of the arguments put forward by the Authors and understand the urgency to conduct this research as per the original protocol and within the limits posed by the resources available. I believe that the manuscript would be strengthened if the Authors incorporated some of the arguments from their latest response into the methods and the discussion section of the manuscript, i.e. to offer an additional justification (and a discussion of the limitations) for the different decisions made regarding study design.

7. PLOS authors have the option to publish the peer review history of their article (what does this mean?). If published, this will include your full peer review and any attached files.

Reviewer #4: No

Reviewer #5: No

---

## [Author Response · Author response to Decision Letter 4]

9 Jun 2021

Response to Reviewers

PONE-D-20-05561R3 

The effect of mobile text messages on knowledge and perception towards cancer and behavioral risks among college students, Northeast Ethiopia: a randomized controlled trial protocol

PLOS ONE

On behalf of all authors, I would like to thank the editor and reviewer for their valuable and constructive comments to improve the excellence of this paper. We revised the manuscript based on the comments. Here is a point by point response for each comment and questions.

Editor

As you will see from the reviewer comments below, only very minor further revisions are requested, namely 1) to change minor inaccuracies in the manuscript and 2) to add some of the justifications provided in the responses to reviewers regarding decisions on study design in the actual manuscript.

Response: Thank you for your suggestions. We addressed all comments and looking forward your prompt response.

Reviewer # 4

The authors have done a fine job addressing the reviewer comments. I only have two remaining points. 

Response: Thank you

 In the abstract, "To test the change in score of HBM" is unclear and should be changed to "To test the change in knowledge and perception" or similar. 

Response: Thank you for your suggestions. We modified it in line 44 of the revised manuscript. 

The 'Trial status' section still says that "We are currently recruiting and enrolling students in the study. As of February 2020, we have enrolled 160 participants." Given the authors' response that the intervention is yet to be developed and delivered, please amend this accordingly.

Response:

Thank you for your suggestion. We amended it in line 308. 

Reviewer #5

I would like to thank the Authors for responding in detail to my suggestions and questions. It is very helpful to see it clarified that the primary outcome are changes in knowledge/perceptions, rather than behaviours. I accept most of the arguments put forward by the Authors and understand the urgency to conduct this research as per the original protocol and within the limits posed by the resources available. 

Response: Thank you 

I believe that the manuscript would be strengthened if the Authors incorporated some of the arguments from their latest response into the methods and the discussion section of the manuscript, i.e. to offer an additional justification (and a discussion of the limitations) for the different decisions made regarding study design.

Response:

Thank you for your suggestion. We incorporated major arguments which could improved the excellence of the paper in methods section from lines 121 to 122, and the discussion section in lines 284, 285, and 303 to 306.

---

## [Editor Report · Decision Letter 5]

15 Jun 2021

The effect of mobile text messages on knowledge and perception towards cancer and behavioral risks among college students, Northeast Ethiopia: a randomized controlled trial protocol

PONE-D-20-05561R5

Dear Dr. Hussien,

We’re pleased to inform you that your manuscript has been judged scientifically suitable for publication and will be formally accepted for publication once it meets all outstanding technical requirements.

Kind regards,

Lion Shahab, MA MSc MSc PhD CPsychol

Academic Editor

PLOS ONE
---

## [Editor Report · Acceptance letter]

30 Jun 2021

PONE-D-20-05561R5 

The effect of mobile text messages on knowledge and perception towards cancer and behavioral risks among college students, Northeast Ethiopia: a randomized controlled trial protocol 

Dear Dr. Hussien:

I'm pleased to inform you that your manuscript has been deemed suitable for publication in PLOS ONE. Congratulations! Your manuscript is now with our production department. 

Kind regards, 

on behalf of

Dr. Lion Shahab 

Academic Editor

PLOS ONE